# Shallow Crustal Structure of S-Wave Velocities in the Coastal Area of South China Constrained by Receiver Function Amplitudes

**Xin Zhang [1], Yinping Qian [1], Xuzhang Shen [2,\*], He Huang [2] and Haibin Chai [3]**

[1] Guangdong Earthquake Agency, Guangzhou 510070, China; zxdqwl@163.com (X.Z.); yinpingqian@163.com (Y.Q.)

[2] School of Earth Science and Engineering, Sun Yat-Sen University, Zhuhai 519000, China; huangh365@mail.sysu.edu.cn

[3] Fourth Habour Engineering Investigation and Design Institute of the Ministry of Transport, China Communications Construction Company Limited, Guangzhou 510000, China; chaihb@fhdigz.com

\* Correspondence: shenxzh5@mail.sysu.edu.cn

**Abstract:** As a traditional method, passive seismic exploration is used to construct the body-wave velocity structure of the upper crust, but it is cost-ineffective and depth-limited when applied to large areas. In this study, we use another more economical method to determine the S-wave velocity (SWV) of the upper crust based on the principle that the amplitude of the direct P-wave on the teleseismic receiver function is sensitive to the upper crust. Using the amplitudes of the massive receiver functions from permanent broadband seismic stations, the SWV structure of the upper crust is obtained in the coastal area of South China (CASC). A pattern of high to low SWVs is exhibited across the study area, with SWVs varying about 2.5–3.7 km/s from west to east. In the profile parallel to the coastline, lateral variations in the SWV correspond to the fault zone, indicating that the cutting depth of most coastal faults is approximately 10 km. Referring to previous studies, we deduce that the low SWV in most sub-areas can be interpreted as the joint effect of the sedimentary layer of the alluvial plain and the accumulation of underground heat flows, in addition to multistage fracturing tectonism. Moreover, the gradual change in the SWV in each profile from the surface to approximately 10 km is correlated with multiple invasions and the coverage of volcanic rocks, to a certain extent.

**Keywords:** S-wave velocity (SWV); multistage intrusive rocks; coastal area of South China; amplitude information of the receiver function

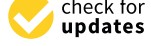



## 1. Introduction

The coastal area of South China (CASC) is near the northern margin of the South China Sea and typically has small earthquakes. Due to complex tectonic activity between the Cathaysia block and the Pacific plate, this area is important for the eastern margin structural framework [1,2]. Faults in the CASC can be divided into three groups, namely, NNE-NE-trending, EW-ENE-trending and NW-WNW-trending, in which the NNE-NE-trending faults have the most obvious tectonic features [3,4]. Currently, some faults exhibit strong activity and remain important earthquake locations. Locations with seismicity depend on geological tectonic activity processes, crustal structures and material properties. The bottom of the upper crust in the CASC is remarkably undulating, lies at depths of approximately 10 km under the Yangtze block and 6–7 km under the Cathaysia block [5]. Zhao et al. [6] and Ye et al. [7] suggested that the average lithology of the upper crust in both blocks is granite-granodiorite and biotite gneiss. The granite-granodiorite layer beneath the Cathaysia block is much thicker. Abundant geophysical data [8] have revealed that the Yangtze and Cathaysia blocks have crustal structures that are characterized by an average composition of granite gneiss for the upper crust, with the presence of mica quartz

schist, felsic granulite, paragranulite, and granite-granodiorite beneath the Yangtze block. To explore the upper crust, ambient noise imaging and artificial source seismic methods are the main strategies for inspecting large-scale material compositions and Cathaysia block-to-sea crust transitions [9–13]. However, the S-wave velocity (SWV) of the upper crust in the CASC, including the constraint of the receiver function method, still needs to be explored in detail.

As one of the most effective seismologic methods, the receiver function is used to study the crustal thicknesses and velocity structures under seismic stations. In the traditional process, the arrival time of the seismic phase is easy to measure in a relatively stable manner. Various studies have obtained the velocity structure based on the arrival time [14–17], while the H-K superposition is a classic strategy for utilizing the temporal information of the receiver function. Moreover, the amplitude of the receiver function can also be adapted to obtain the SWV of the upper crust, as well as the velocity and density contrasts of the Moho discontinuity. Referring to previous studies on receiver function amplitudes [18–20], Qian et al. [21] first proposed this idea of data processing, and developed a method to inverse the SWV structure of the upper crust with the amplitude of the receiver function [22,23].

The study area exhibits episodic tectonism between the Pacific and Eurasian plates [24,25], where the outcrops of late Mesozoic igneous rocks extend beyond the boundary of the CASC block. The CASC was the transitional zone between the Cathaysian block and oceanic crust, and was influenced by the subduction of the Pacific plate. The back-arc faulting and seafloor expansion of the CASC were associated with multiple tectonic movements since the Cretaceous [26,27]. A large number of Mesozoic intrusive rocks developed along these NE-trending faults. The NW-trending faults formed later than the Cretaceous intrusive events, resulting in strong neotectonics [26]. However, the magmatic suite that intruded in the Mesozoic had a mantle source composed of late Archean and Mesoproterozoic crustal components of xenoliths. Moreover, as a result of collisions and regional metamorphism, the volcanic suite and sediments were altered, forming abundant metamorphic rocks in the Lianhuashan fault (LHSF) zone. Moreover, heat flows and strong metamorphic activity are manifested by dozens of faults. Many strong earthquakes, such as the largest reservoir-induced earthquake, recorded as Ms 6.1 in the Xinfengjiang reservoir (XFJR), and the largest oceanic earthquake, recorded as Ms 7.2 near Nan'ao Island, have caused great damage to the CASC. In order to elucidate the heat flows and seismogenic structure, we use the amplitude of the receiver function to obtain the SWV structure of the upper crust. By comparing our findings with previous studies, we attempt to establish a model for the evolution of the structural environment.

## 2. Data and Methods

### 2.1. Geological Setting

The distribution of seismic stations covers the whole CASC, as shown in Figure 1a. The faults in Figure 1c mainly have NE-SW strikes, of which the typical faults are the LHSF, coastal faults, the Heyuan fault, and the Xijiang fault (XIJF). The XFJR has the characteristics of frequent occurrences of small earthquakes and dense stations, and its seismogenic structures are evaluated later.

### 2.2. Method of the Direct Wave Amplitude of the Receiver Function

In this study, seismic waveform data from 68 broadband stations in the CASC during 2007–2011 are used to calculate the receiver function. The seismic data recorded on three-component digital broadband stations. These stations were equipped with 24-digit broadband/very broadband seismometers (60 s or 120 s) with a sampling rate of 100 per second. Teleseismic events from the waveform data are intercepted at Ms > 5.5, with epicentral distances ranging from 30° to 90°. The distributions of seismic events used in the receiver function calculation are shown in Figure 1b. Each teleseismic event in the

30°–90° range can be assumed to be an incident wave with a steep angle, and the wave likely propagated to the station as a plane wave.

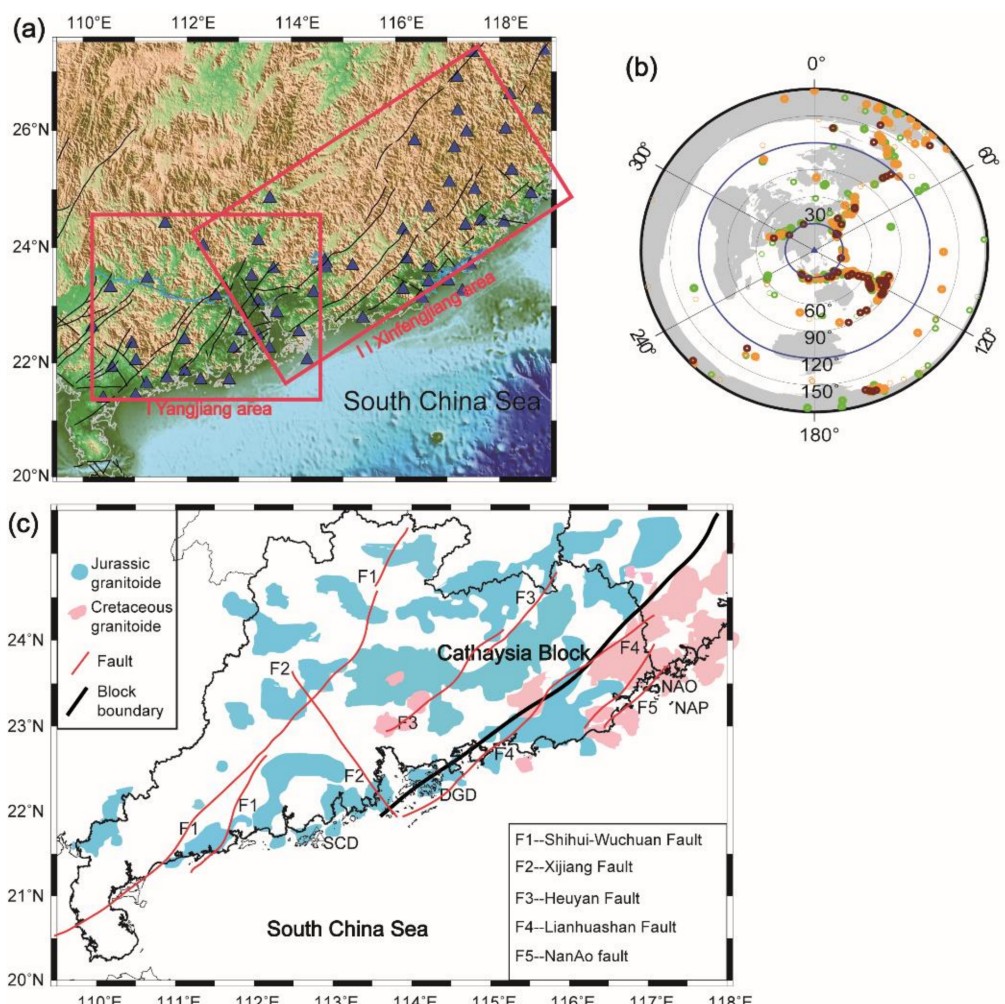

**Figure 1.** Spatial distributions of seismic stations (**a**) and seismic events (**b**) in the study area. The seismic events that are used in calculating the receiver function are shown in (**b**). The distribution of Mesozoic granites in (**c**) was retrieved from Li [28], while the late Mesozoic volcanic arcs were retrieved from Zhou et al. [3] and Li et al. [29]. LHSF represents the Lianghuashan fault; SWCF represents the Sihui-Wuchuan fault. XIJF and HEYF represent the Xijiang fault and Heyuan fault, respectively.

As the Pp arrives at the seismic station, it forms a direct P-wave (Pp) or is converted into an S-wave (Ps) at the interface. It also forms multiple waves when reflected at the discontinuities below the surface. In such multiple reflections, not only is conversion between the P-wave and S-wave adopted, but PpPs and PpSs + PsPs combinations are also generated.

When the P-wave spreads through the 1-D velocity model with horizontally layered structures, it produces a ground displacement response [14]. The displacement response can be expressed in the time domain as:

$$Z(t) = \sum_{k=0}^{n} z_k s(t - t_k) \tag{1}$$

$$R(t) = \sum_{k=0}^{n} r_k s(t - t_k) \tag{2}$$

where Z(t) and R(t) represent displacements of the vertical and radial components of motion, respectively; $s(t)$ represents the time function of the effective source; $t_k$ represents the arrival time of the seismic phase, where $k = 0$ indicates the direct P-wave; and $z_k$ and $r_k$ represent the amplitudes of the vertical and radial components of the seismic phase, respectively, and represent the values of the reflection coefficients obtained by multiplying the transmission coefficients. Through deconvolution of the horizontal component by the vertical component of the surface displacement, the medium structural response in the frequency domain is expressed as the following:

$$H(\omega) = \frac{S(\omega)R(\omega)}{S(\omega)Z(\omega)} = \frac{R(\omega)}{Z(\omega)} \tag{3}$$

where $\omega$ represents the ray frequency, $S(\omega)$ represents the source spectrum, and $H(\omega)$ represents the medium structural response and radial receiver function. Through Fourier transformation of the receiver function, the following are obtained:

$$R(\omega) = r_0 \sum_{k=0}^{n} \hat{r}_k e^{-iwt_k} \tag{4}$$

$$Z(\omega) = z_0 \sum_{k=0}^{n} \hat{z}_k e^{-iwt_k} \tag{5}$$

where $\hat{z}_k$ represents the k-phase amplitude after processing the direct P-wave pulse amplitude in relation to the vertical component; that is, $\hat{z}_k = z_k/z_0$ and $\hat{r}_k = r_k/r_0$.

The receiver function indicates seismic properties; therefore, the discontinuity structure can be deduced by calculating the radial receiver function. Given the simple model of the horizontal single layer, there are only direct P-waves, P-S converted waves and resonant multiple waves; thus, Equations (4) and (5) hold and can be simplified as follows:

$$R(\omega) = r_0 \left[ 1 + \hat{r}_p e^{-iwt_p} + \hat{r}_s e^{-iwt_s} \right] \tag{6}$$

$$Z(\omega) = z_0 \left[ 1 + \hat{z}_p e^{-iwt_p} + \hat{z}_s e^{-iwt_s} \right] \tag{7}$$

Substituting (6) and (7) into (3) and ignoring the higher-order terms, the following is obtained:

$$H(\omega) = \frac{r_0}{z_0} \frac{1 + \hat{r}_s e^{-iwt_s} + 2\hat{z}_p coswt_p}{1 + 2\hat{z}_p coswt_p} = \frac{r_0}{z_0} \left[ 1 + \hat{r}_s e^{-iwt_s} \right] \tag{8}$$

Equation (8) shows the absolute amplitude of the direct P-wave on the free surface ($t = 0$). The amplitude ratio, $r_0/z_0$, denotes the velocity structure near the surface. It is very important to add the amplitude ratio information into the receiver function. Assume a constant velocity near the surface. We can combine the equations for the free-surface displacement due to an incident P wave [30]:

$$\frac{r_0}{z_0} = \frac{2p\eta_{V_{s0}}}{Vs_0^{-2} - 2p^2} \tag{9}$$

where $\eta_{V_{s0}} = \left( 1/(Vs_0)^2 - p^2 \right)^{1/2}$ represents the vertical slowness of the S-wave and p represents the horizontal slowness (i.e., ray parameter), while $Vs_0$ represents the SWV near the surface. Equation (9) shows the relationship between the amplitude and average velocity at low frequencies. For higher-frequency bands, amplitudes of the seismic phase must be calculated using synthetic seismograms; however, the direct P-wave amplitude ratio is dependent on the near-surface velocity [31].

The direct P-wave amplitude is correlated only with the SWV ($Vs_0$) and ray parameter (p) near the surface. The ray parameter can be calculated from the epicentral distance and focal depth [32]. When the epicentral distance is provided, the direct P-wave amplitude is mainly affected by the SWV near the surface. Based on the amplitude value of the P-wave receiver function, the SWV of the upper crust can be calculated. Since the receiver function is associated only with SWVs under seismic stations, the receiver functions of different epicentral distances are similar for stations with uniform transverse structures of the crust. Therefore, the amplitudes of the direct P-waves at different epicentral distances at the same station can be applied in order to constrain the SWV of the upper crust.

### 2.3. Method of the SWV

In order to verify the feasibility of the actual data processing process, we compare the theoretical seismogram [33] with the actual data processing results. First, we establish a single-layer crustal velocity model (Figure 2a). The SWV of the crust is 3.75 km/s, while the P-S velocity ratio is 1.732. The velocity structure under the crust refers to the IASP91 model [34]. Second, we select an epicentral distance of 70° and calculate the theoretical seismogram by the reflection transmissivity method [33]. Then, we select the Gaussian filter factor (alpha = 1.5) and calculate the theoretical receiver function using the time domain iterative deconvolution method [35]. Finally, we measure the Pp phase amplitude in the theoretical receiver function. The initial movement of the P-wave is taken as the starting time. The maximum value of the wave peak near the starting time (±2 s) is the P-wave amplitude.

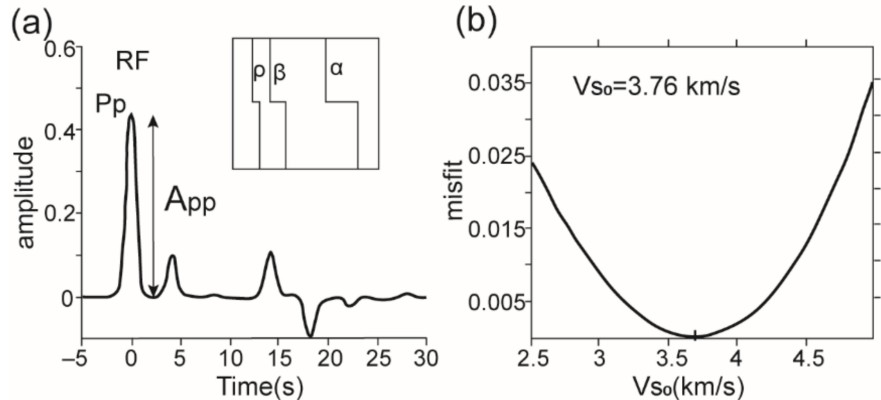

**Figure 2.** Receiver function of a single epicentral distance (**a**) and the subsurface SWV (**b**).

We use the grid search method and the Pp-wave amplitude to determine the SWV. The values are 2.5–3.5 km/s, while the search step is 0.01 km/s. For each search, the theoretical value of the absolute Pp-wave amplitude can be calculated by using Equation (9). In the search process, the theoretical value of the Pp-wave amplitude with different epicentral distances can be calculated by the $Vs_0$ in each step. The difference between the observed and theoretical values can be expressed as the objective function, $\text{misfit}(Vs_0)$ :

$$\text{misfit}(Vs_0) = \sqrt{\sum_i \left( A_i^{\text{obs}} - A_i^{\text{syn}} \right)^2} \tag{10}$$

where $A^{\text{obs}}$ represents the observed value of the Pp amplitude, $A^{\text{syn}}$ represents the theoretical value of the Pp amplitude, and $\Sigma$ is the grid scanning superposition of different ray parameters. The optimal value can be obtained by calculating the misfit ($Vs_0$) according to Equation (10) and taking the horizontal coordinate corresponding to the minimum value. The Pp-wave amplitude in the R-direction receiver function can be measured in order to obtain the $Vs_0$ (Figure 2a).

$A_{Pp}$ represents the direct wave (Pp) amplitude of the receiver function, while the "+" symbol represents the optimal $Vs_0$ to be searched in the abscissa.

### 2.4. Depth and Alpha

We use a Gaussian filter ($\exp(-\frac{\omega^2}{4a^2})$) to suppress the high-frequency noise when extracting the receiver function. The effective cutoff frequency is defined as the frequency, while the filter amplitude is reduced to its $\frac{1}{e}$.

Without depth information while calculating $Vs_0$ based on Equation (9), we cannot directly determine its specific depth. In order to test the depth distribution range of SWV, we tested four models in Figure 3. In Figure 3a,c, the velocity above the 35 km discontinuity increases in gradient, in which the difference between them is that the increasing trend is different; the growth is faster and the value on the surface is greater in Figure 3a. In Figure 3b,d, the depths of the two discontinuities are 20 km and 35 km, respectively, and the velocity remains unchanged below 35 km and increases linearly above 20 km; the difference between them is that the increasing trend is different; the growth is faster and the value on the surface is greater in Figure 3b. Figure 3e shows the SWV corresponding to multiple Gaussian filter factors obtained by our method. When the Gaussian filter factor becomes larger (meaning shorter wavelength), the corresponding S-wave velocity becomes smaller, and the depth corresponding to the S-wave velocity gradually becomes shallower. When comparing the calculated SWV with the average SWV in these models, the calculated value is close to the average value of a certain depth of the model, reflecting the depth that corresponds to a filter factor (as shown in Figure 3e). With this processing approach, we select different filter factors in order to calculate the receiving function, and then obtain the conversion of SWV in depth.

Gaussian filtering is the classical smoothing method in the deconvolution process of receiver functions. This study's method essentially uses the dispersion of receiver function amplitude. The relationship between alpha and the depth of SWV has no quantitative result in mathematics, so we finally give the most sensitive depth of SWV based on reasonable numerical experiments (Figure 3). When alpha is smaller, it indicates the average value of SWV from the surface to the deeper crust; when alpha is larger, it indicates the average value of SWV from the surface to the shallower crust (Figure 3). Wang et al. [22] showed the mode in which different alpha factors correspond to the SWV at a certain depth in the upper crust by calculating the amplitude of the receiver function. This method involves the following two steps: first, the high-frequency approximate amplitude formula of direct P-waves in receiver functions of individual stations is used to fit the observed amplitude distribution against the ray parameters at different frequencies; and second, the S-wave velocity depth profile beneath each station is constrained according to an empirical correlation between frequency and depth. In this study, we referred to the results reported by Wang et al. [22,23].

There are divergences in receiver function amplitudes corresponding to different alpha factors in Figure 3. In order to test the influence of different Gaussian factors (alpha), we choose the values of 1.0, 1.5, 2.5, 3 and 5. Based on the model shown in Figure 3a–d, we use two groups of receiver functions to detect the influence of different filter frequency windows on the receiver function amplitude. Each receiver function for each epicentral distance can yield a $Vs_0$; thus the optimal $Vs_0$, which is determined by all receiver functions, is used. The least squares fit error ($\varepsilon$) is calculated by the following:

$$\varepsilon = S_y \sqrt{\frac{1}{N} \frac{\left(V_{s0}(i) - \overline{V}_{s0}\right)^2}{\sum_{i=1}^{i=N}\left(V_{s0}(i) - \overline{V}_{s0}\right)^2}} \tag{11}$$

where $N$ represents the number of waveforms with different epicentral distances at each station; $\overline{V}_{s0}$ represents the average value of all $V_{s0}(i)$ values; $S_y = \sqrt{U/(N-2)}$ represents the residual mean square deviation; U represents the sum of the regression squares of different $V_{s0}$ values; and the error percentage is $\delta = \epsilon / \overline{V}_{s0} \cdot 100\%$.

In general, a larger alpha indicates a shallower SWV (Figure 3e). For example, alpha = 1.5 shows that the SWV is at a depth of approximately 5 km; alpha = 6 indicates that at a depth of 1 km, the SWV is near the surface; and alpha = 1 shows that the SWV is at a depth of 8 km in the upper crust. Although four velocity-density models are used in the test process, the trend of alpha to the $Vs_0$ depth is similar. When alpha < 2, the results tend to be consistent, which means that the velocity structure of the S-wave is more reliable below 4 km. According to the numerical test results, different *alphas* indicate S-wave velocities at different depths; therefore, the comprehensive results can show the SWV structure of the upper crust.

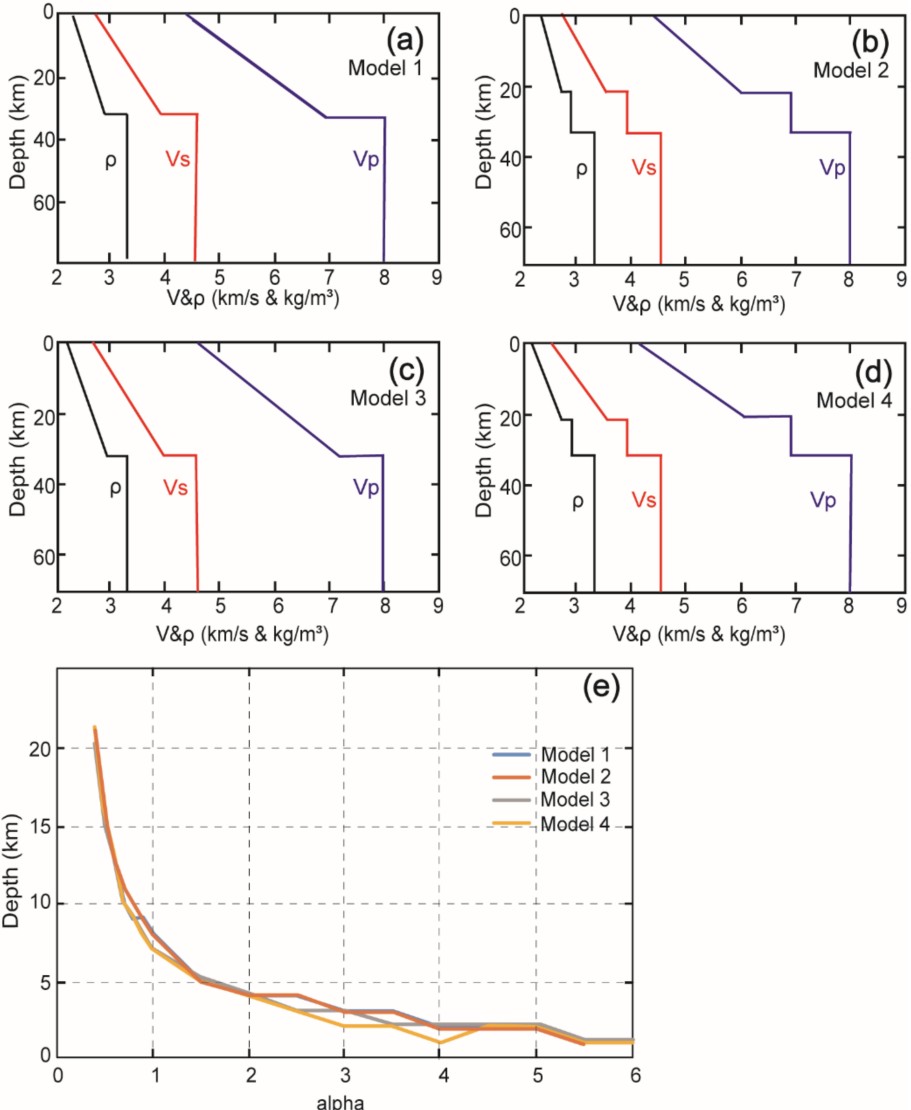

**Figure 3.** Gaussian filtering factors (alpha) corresponding to SWVs at different depths. (**a**–**d**) Crustal structure model with different velocity-density values; (**e**) test results of the models.

### 2.5. Data Processing Procedures

Our data processing procedures consist of the following 5 steps:

(i) Events with epicentral distances of 30° to 90° and M ≥ 5.5 are selected from the continuous waveform data. Waveforms with high signal-to-noise ratios and clear initial motions are extracted and preprocessed for filtering, deburring, mean value removal, and linear trend removal, among others.

(ii) Based on the earthquake azimuth, the waveform from the N-E-Z component is rotated to the R-T-Z component. Then, the data from 20 s before to 150 s after the initial P-wave

movement are intercepted. Gaussian filter factors are selected (such as alpha = 1.0, 1.5, 2.5, 3, 5, and 7) in order to filter the signal of all stations. The receiver function is calculated by the time domain deconvolution method and filtering process.

(iii) Using the vertical component of the receiver function to deconvolute itself, its maximum amplitude value is obtained to unify the radial receiver function, in order to obtain the absolute amplitude [15].

(iv) Direct wave amplitudes of receiver functions at different stations in different directions are measured as observation values ($A^{obs}$). For each $Vs_0$, the theoretical value ($A^{syn}$) of the direct wave amplitude is calculated using Equation (1). The smallest value of the misfit ($Vs_0$) corresponds to the optimal solution of the $Vs_0$. Finally, the SWV near the surface of the area is obtained.

(v) In order to improve the signal-to-noise ratio, receiver functions of different epicentral distances are evenly distributed in 60° epicentral distance units, while receiver functions in each unit are superposed into every 1° grid size. Figure 4a shows the superposition results of multiple receiver functions at different epicentral distances obtained by the DNB station, with a high signal-to-noise ratio and clear seismic phases. Figure 4c shows the SWV $Vs_0 = 3.21 \pm 0.22$ km/s of the DNB station, with a percentage error of δ = 6.8%. Figure 4b,d show the superposition results of multiple receiver functions at different epicentral distances from the YGX station, with $Vs_0 = 2.78 \pm 0.21$ km/s and δ = 7.5%.

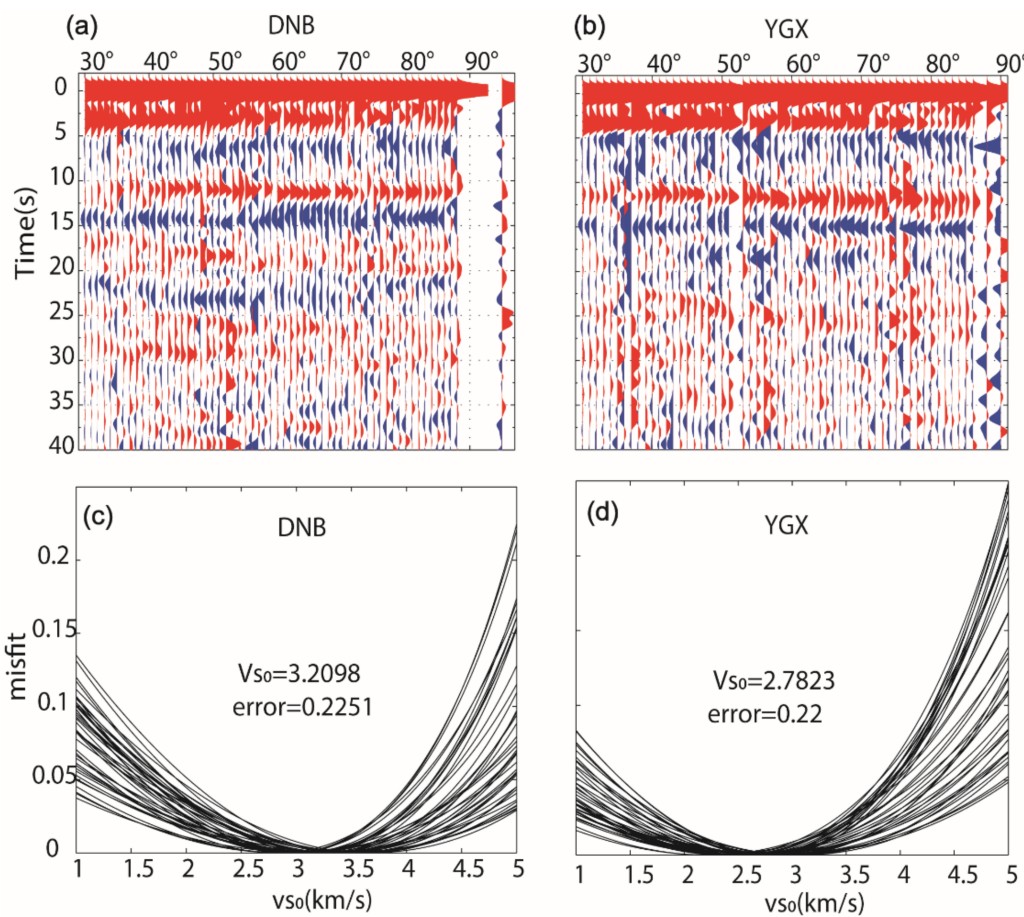

**Figure 4.** Receiver functions of the DNB and YGX stations (**a**,**b**) as well as their subsurface SWVs (**c**,**d**). Receiver functions of different epicentral distances to the DNB and YGX stations (alpha = 1.5). The thin line indicates the $Vs_0$ search process, while the "+" symbol represents the optimal $Vs_0$ to be searched in the abscissa.

## 3. Results

### 3.1. Spatial Distribution of the SWV

According to the epicentral distance, the receiver functions of each station are stacked in each 1° range; therefore, the SWV is the mean value of the area adjacent to each station. In Figure 5, the SWVs of the upper crust in the CASC are 2.6–3.6 km/s. From west to east, the upper crust of the CASC can be divided into four regions from high- to low-SWV trends, corresponding to the distribution of intrusive rocks in the four periods. In the northeastern part of the study area, the highest value is 3.6 km/s, which is much higher than those in the central areas (2.5–2.8 km/s). The northern part (ZHJ-CTTZ station) is also part of the low-SWV region, which is less than 2.8 km/s. The SWVs of the western area are generally 2.8–3.5 km/s, and are lower in the south and higher in the north, which may be correlated with widespread underground thermal activities and intrusive rocks.

The SWV of the area containing faults extending in the NW-SE direction should be higher than that of other areas, with a value greater than 3.2 km/s. These faults thicken in Figure 6, and are located in the adjacent areas of the DNB station, YND station, and YSCG station. Speculating that the fault zone in this direction may have formed high SWVs and was not damaged by intrusive rocks is reasonable. This phenomenon is caused by the NW-trending faults that formed later than the Cretaceous intrusive events, and induced strong neotectonics [26].

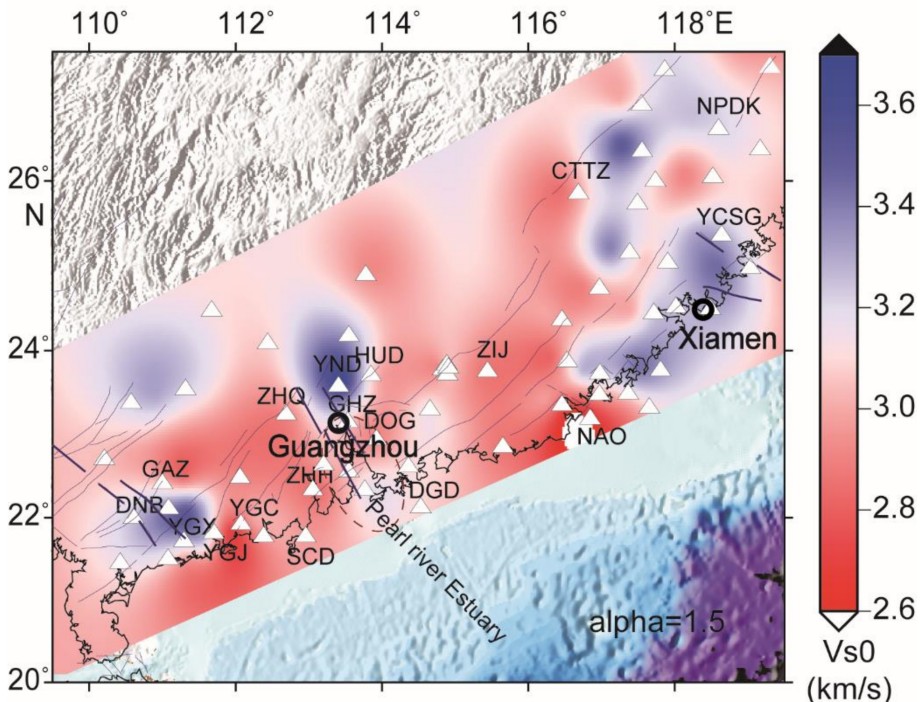

**Figure 5.** SWVs after the interpolation of each station (alpha = 1.5, H ≅ 5 km) around the study area. The white triangle represents the seismic station. The tight line represents the fault, and the thick line represents the fault extending in the NW-SE direction.

### 3.2. SWV Distribution with Depth

The SWV exhibits an increasing trend with depth in most areas from 1 km to 8 km (in Figures 6 and 7), corresponding to the loose geological sedimentary layer, loose locally weathered rock layer, and dense bedrock layer. In order to analyze the longitudinal SWV structure, Figure 6 shows longitudinal slices of the SWV structure in the XFJR and its adjacent areas. The western XFJR is a high-velocity area, whose range extends to 8 km underground, whereas the eastern XFJR is a low-velocity area. The phenomenon that the SWV in the center (~5 km) is lower than that in the shallow zone (>1 km) and in the deeper

zone (~11 km) indicates that heat flows and strong rheological effects exist in the deep zone. However, a high- to low-velocity transition area appears in the eastern XFJR, in which the surface velocity (~1 km) is significantly lower than that in the deep zone (~8 km). It may be affected by the impoundment of the reservoir, resulting in the shallow layer being more broken and filled by the fluid.

Figure 7 shows the longitudinal slice of the SWV structure in Yangjiang and its adjacent areas. In the horizontal direction, the western DNB station is an obvious high-velocity area, whereas the southern area is a relatively low-velocity area. The SWV in the coastal area is higher than that in the inland area and is relatively stable in the longitudinal direction. The lateral variation in the SWV in the deep layer is smaller, while the lateral variation in the shallow layer is more complex than that in the deep zone.

As shown in Figure 8, profile AA' is the ZHJ-YCSG station, and profile BB' is the YLS-NPDK station, which represents the penetration depth of the fault zone. In profile AA' (Figure 8b), the SWV under the HEY station has an obvious lateral difference, while the relatively high-velocity body with an overall eastward dip extends from the lower crust to the surface. The same trend appears in profile BB' (Figure 8a), except that the lower SWV layer becomes shallower (from 10 to 7 km).

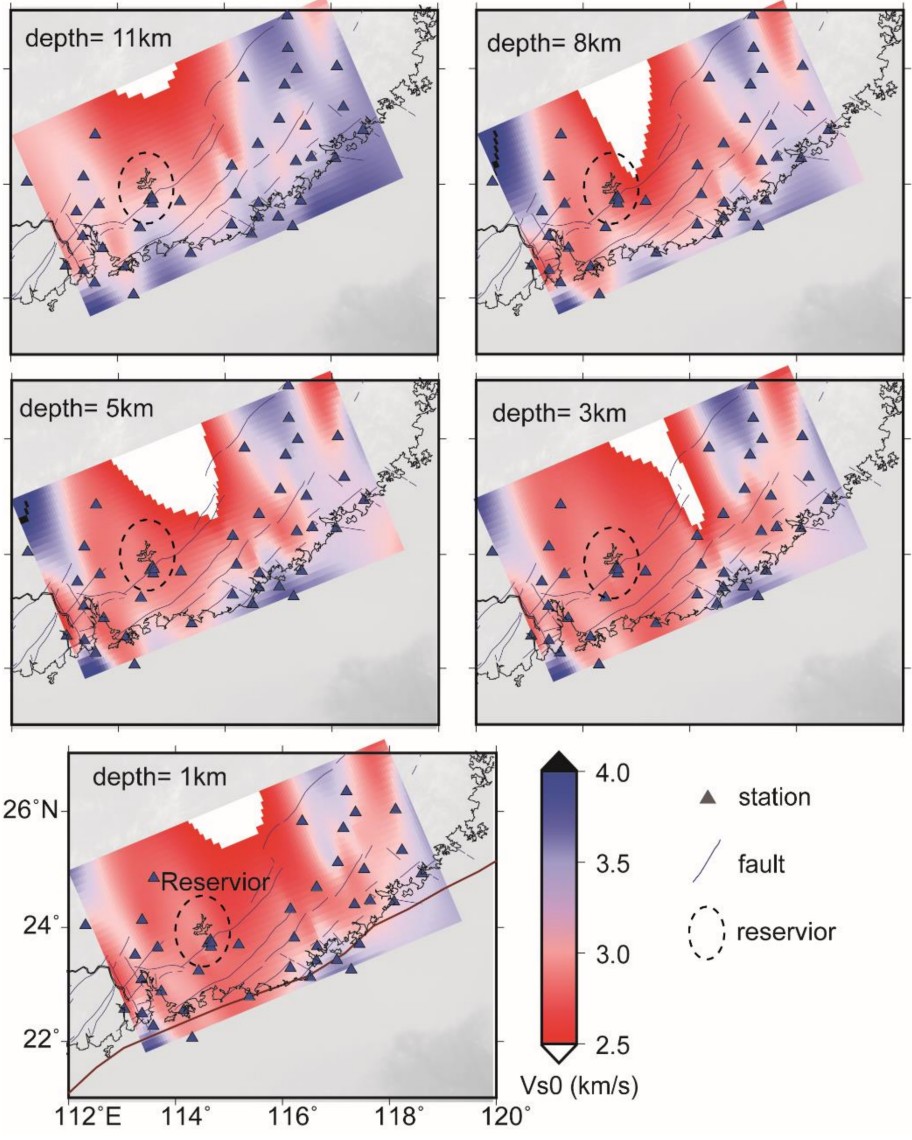

**Figure 6.** SWV tomographic structure in the XFJR and its adjacent area. The depth range of 11 to 1 km corresponds to alpha values of 0.5 to 5.

Generally, the fault zone contains a transition zone between high and low SWVs, and the penetration depths of most faults reach the lower crust, such as that of the SHWF (Figure 8b). However, the penetration depth of the LHSF is controversial. Geological studies [28,29] show that the Upper Jurassic group, which is developed in the LHSF zone, is a continental intermediate–acidic volcanic rock modified by dynamic metamorphism, with thicknesses of 1284–6419 m, which is equivalent to the depth of the shallow low- to high-velocity body transformation zone in Profile AA'. In addition, the low-velocity area (over 5 km) on the east side of the LHSF may be the area in which heat flows are concentrated, which softens the formation as a result of concentrated thermal subsidence. In addition, the locations of XIJF and HEYF are clear transformation zones of low to high velocities in profile BB'. The depths of the two faults are less than 5 km, and the high-velocity layer above 3.4 km/s does not appear below 9 km.

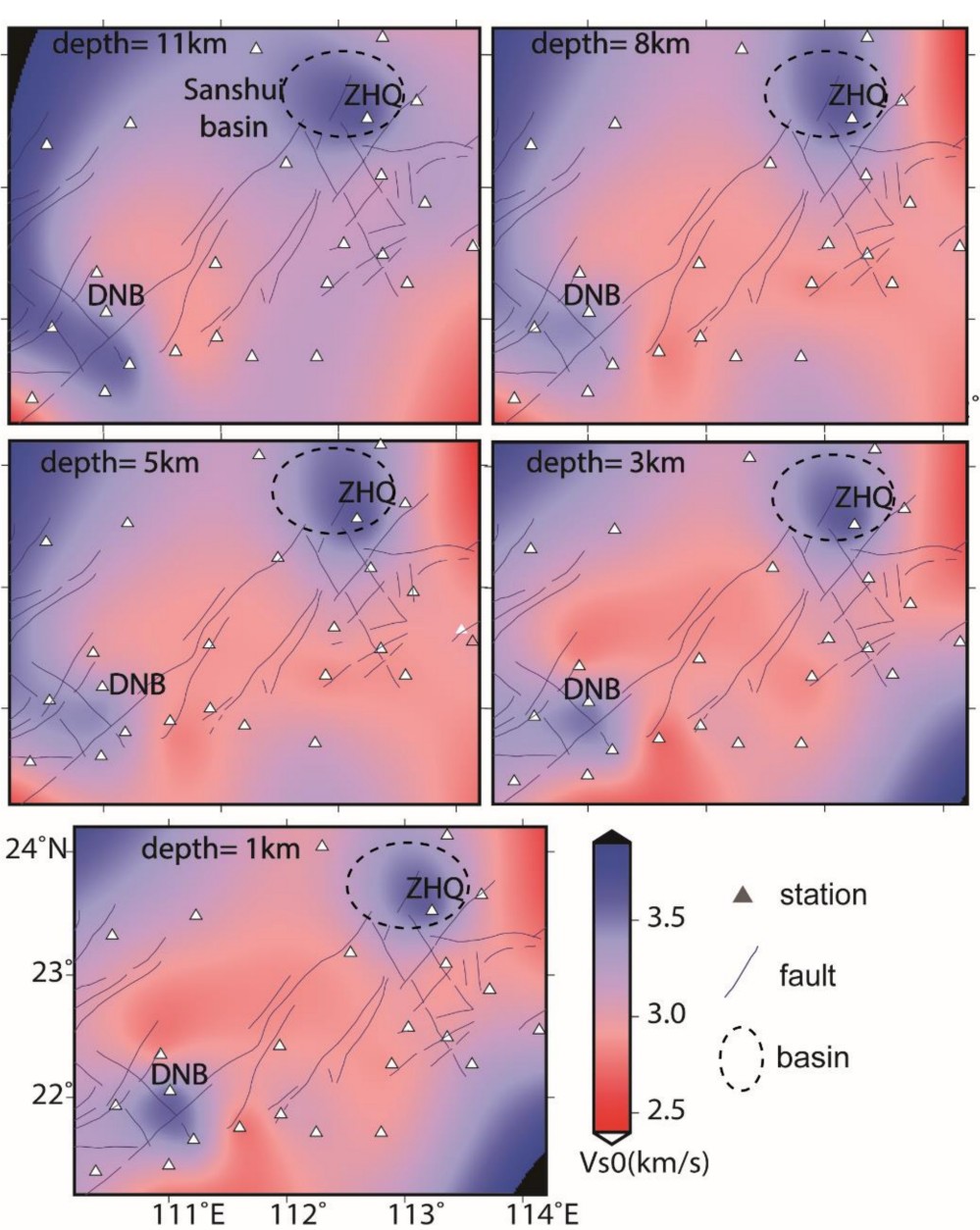

**Figure 7.** SWV tomographic structure in Yangjiang and its adjacent area.

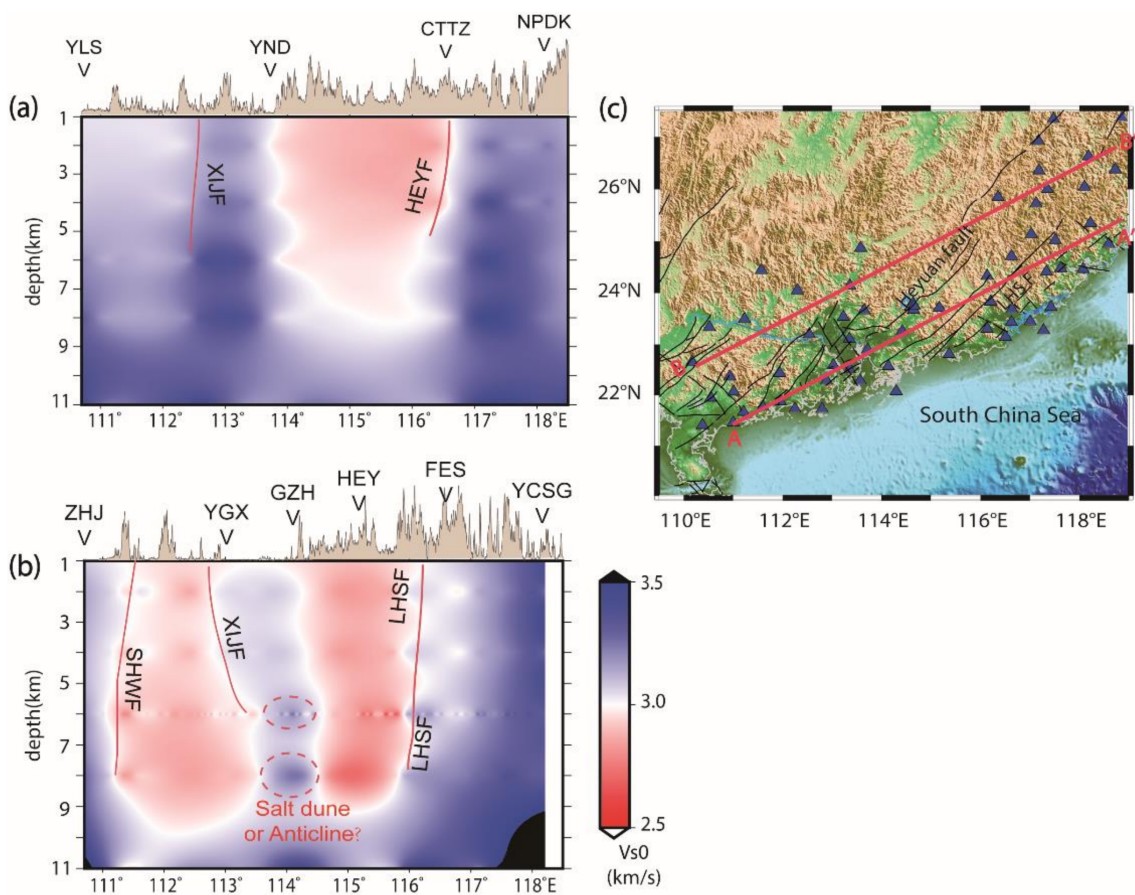

**Figure 8.** SWV structure profile parallel to the coastline. (**a**) SWV structure of the YLS-NPDK profile (BB′); (**b**) SWV structure of the ZHJ-YCSG profile (AA′); (**c**) locations of the two profiles. LHSF represents the Lianhuashan fault; SWCF represents the Sihui-Wuchuan fault. XIJF and HEYF represent the Xijiang fault and Heyuan fault, respectively.

## 4. Discussion

Compared with previous findings, some overlaps were present between our study area and those of Zhang et al. [36], Shen et al. [37] and Huang et al. [38]. According to the Rayleigh wave sensitivity depth curve [36], the velocity group within the 2–8 s period reflects the SWV structure at 10 km or deeper. In particular, the velocity group is near the 5 s period, which is most sensitive to a depth of 5 km. Figure 9a shows the mean SWV value with alpha = 1.5 in the central area, which is close to that of the Rayleigh wave velocity group with a period of 5 s. The similarity of the two results is attributed to the switch between high and low velocities, indicating that the SWV structure in the upper crust is relatively complex in the transverse direction. In the eastern part of the study area, Zhang et al. [36] reported that the velocity group in the 5 s period was approximately 2.9 km/s, and Huang et al. [38] reported that the velocity group in the 5 s period was approximately 2.8–3.2 km/s (Figure 9a). In this study, the SWVs are approximately 2.9–3.1 km/s, which are comparable with previous results and closer to the results of Huang et al. [38].

A large cluster of exposed intrusive rocks is located in the eastern part of the study area, where the sedimentary layer is very thin in the entire area. However, it is difficult to establish the existence of a thinner sedimentary layer and a low SWV via ambient noise imaging or receiver functions. The most typical example is the NAP station, which is located on a volcanic island over magmatic intrusive rocks; however, its SWV above 8 km is less than 3.0 km/s, which does not correspond with the sedimentary layer thickness. The most reasonable explanation for the low SWV is that an area with seismicity where earthquakes have high activity levels in the middle and lower crust may cause the lithosphere to fracture.

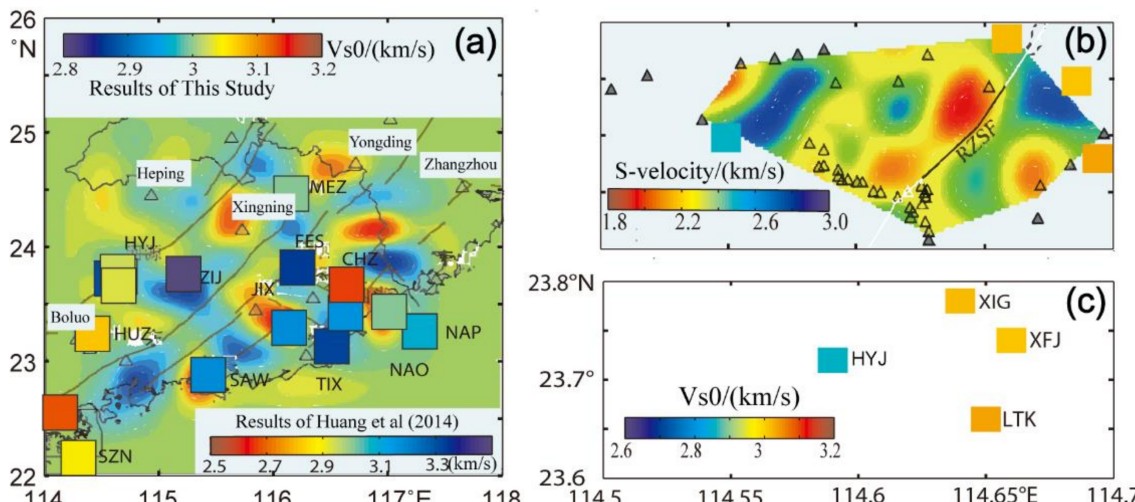

**Figure 9.** Comparisons of SWVs and Rayleigh wave velocity groups of ambient noise tomography in a similar area. The two colored bars in (**a**) are consistent. (**a**) The bottom map is the result of ambient noise tomography by Huang et al. [38] in eastern Guangdong Province; (**b**) the bottom map is a result of ambient noise tomography by He et al. [13] in XFJR; (**c**) finding of $Vs_0$ in this study in the XFJR.

The SWV is a useful parameter for buildings design [39,40], especially in the XFJR area which may experience large earthquakes. Wang et al. [41] reported that SWVs greater than 1 km are less than 3.0 km/s based on ambient noise imaging in a super dense array. In this paper, alpha = 7 shows an SWV near 1 km. In Figure 8b, the SWV (alpha = 7) of the XFJ station is close to 3.0 km/s, which is greater than the noise imaging result; that of the HYJ station is 2.85 km/s, which is close to the noise imaging result; and that of the LTK station is 2.8 km/s, which is close to the noise imaging result on the right side of the HEYF. These results indicate that the SWV near the XFJR surface is uniform in a small zone. However, the SWV in this study is larger than that of the noise imaging of a super dense array. As shown in Figure 8b, the results of noise imaging are 1.6–3.0 km/s, while our findings are approximately 2.9–3.1 km/s (Figure 9c). In addition, since the single station receiver function in this study is the average value of the adjacent area, differences in the S-wave velocities of several stations in the XFJR are very small, but the value is larger than the result obtained by noise imaging. Moreover, the accuracy of the SWV near the shallow surface (~1 km) given by the amplitude information of the receiver function is relatively low and thus is not as good as the noise imaging of the super dense array.

Two main explanations can be given for the differences in S-wave velocities near the surface of the South China Sea. The first is the inverse correlation with the sedimentary layer thickness, resulting in a thicker sedimentary layer indicating a smaller SWV [21,40]. The second is the distribution of the heat flows in the upper crust. High heat flows soften the regional strata, resulting in a low SWV [9]. Therefore, the SWV is higher in the region without heat flows. These two factors may account for the most obvious differences in the SWV structure. However, at a few stations, the results are not consistent with the two explanations. Therefore, other possible reasons, such as the impact of earthquakes in some fault zones, as well as the effects of multiple intrusive rocks in coastal areas, should be considered.

In the Yangjiang area and the Sanshui Basin near the ZHQ station, the thickness of the sedimentary layer indicates a low-SWV area. Under the Sanshui Basin near the ZHQ station, the sedimentary layer is relatively thick, while the SWV is less than 3.0 km/s, and the low-SWV body extends to less than 5 km. In other areas, the low-SWV area is deeper, which may not be due to the thick sedimentary layer and should be analyzed further. On the other hand, the low-SWV area is not always a thick sedimentary layer in the study area. For example, hot springs are exposed areas near FES station, where the SWV is less than

2.8 km/s. Another explanation is that heat flow activity in the upper crust may have led to a low SWV, especially at the FES and PUN stations along the LHSF. However, similar to the western region of the study area, the region with a high SWV contains more hot springs. Although the spatial distributions exhibited obvious regional and zonal characteristics, it is difficult to explain the low SWV bodies in order. In the following study, we will focus on whether the basins come from the invasion and transformation of geological bodies, by establishing more detailed array observation to build 3D crustal structures, as shown in the lithosphere of Italian Central Apennine [42].

The discontinuity of the SWV in the upper crust can indicate the geological structure caused by multistage magma intrusion. Throughout the geological history of the CASC, there have been many stages of intrusive rocks. The high- to low-SWV areas are correlated with the distribution of multistage intrusive rocks, sedimentary layers, structural metamorphic belts, and earthquake fractures caused by intrusive forces (Figure 10).

Since 250 Ma, the CASC has been subjected to several magmatic intrusions [3,4,26] of Triassic granite (250–200 Ma), Jurassic granite (200–150 Ma), Cretaceous granite and Cretaceous volcanic rocks (139–80 Ma), which correspond to multiple subductions and rollbacks of the Paleo-Pacific plate [43]. The previous two intrusions and the latter Cretaceous volcanic rocks are divided by the LHSF (Figure 10). The magmatic intrusion model contains three steps. First, the magma chamber in the mantle upwelled due to the subduction of the Pacific plate. Second, the magma began to break through the Moho surface and the crust, and then it upwelled by utilizing some structural zones as a channel to intrude into the sedimentary rock area. Third, with the strengthened subduction of the Pacific plate, multiple stages of magma pushed and covered each other, forming the present pattern of volcanic rocks, sedimentary rocks, and the contact metamorphic zone (near the fault). In addition, some channels in which magma rose acted as ways for rising mantle heat flows, leading to the current geothermal developments in some areas.

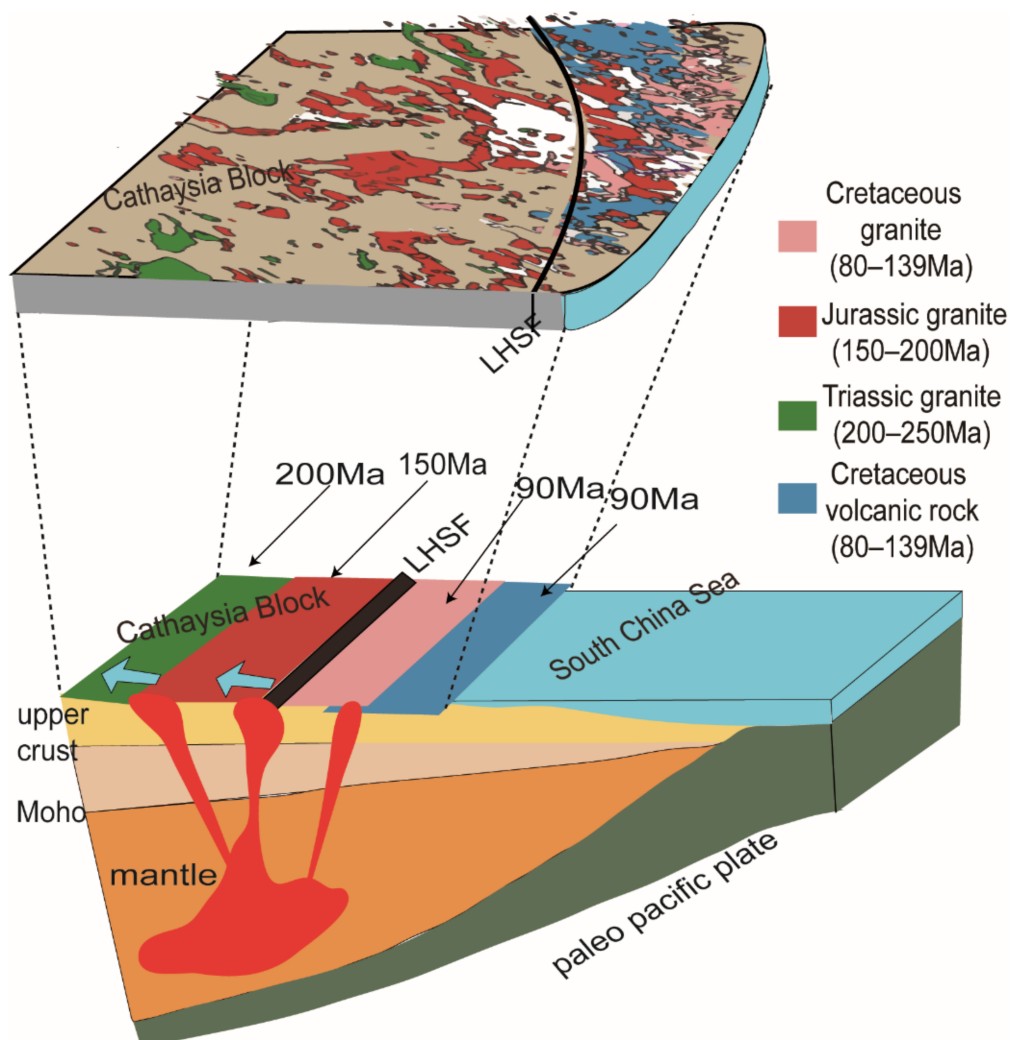

**Figure 10.** Modeling of several intrusive rocks. Since 250 Ma, the CASC has been subjected to three magmatic intrusions: Triassic granite (250–200 Ma), Jurassic granite (200–150 Ma), and Cretaceous granite and Cretaceous volcanic rocks (139–80 Ma).

## 5. Conclusions

Based on amplitude information of receiver functions, the SWV structure of the upper crust over 8 km in the CASC was obtained. In conjunction with the existing geological and petrological research results, SWV structure reveals the intrusion mechanism of multistage subvolcanic rocks near the surface in CASC. We obtained the following conclusions:

(i)    We propose a method of using the amplitude of the receiver function to obtain the SWV that corresponds to the Gaussian filter factor. This method is stable, economical, environmentally friendly and efficient, while comparing with the active source surveying in the upper crust. Meanwhile, the accuracy and separation rate of this method need more numerical experiments and data processing to verify, especially in the area with complex velocity structure.

(ii)    In the study area, we deduce that the low SWV in most sub-areas can be interpreted as the joint effect of the sedimentary layer of the alluvial plain and the accumulation of underground heat flows, in addition to multistage fracturing tectonism. The gradual change in the SWV in each profile from the surface to approximately 10 km is correlated with multiple invasions and the coverage of volcanic rocks to a certain extent, indicating weathering layer coverage and structural transformations between multistage intrusive rocks in the CASC.

**Author Contributions:** Conceptualization, X.S. and Y.Q.; data collecting and processing, H.C., X.Z. and Y.Q.; methodology, X.S. and Y.Q.; writing and original draft, X.Z. and H.H. All authors have read and agreed to the published version of the manuscript.

**Funding:** This study was financially supported by the National Natural Science Foundation of China (Grant 41874052 and 41730212), the Second Tibetan Plateau Scientific Expedition and Research Program (STEP) (2019QZKK0701); project supported by Innovation Group Project of Southern Marine Science and Engineering Guangdong Laboratory (Zhuhai) (No.311021002), Guangdong Province Introduced Innovative R&D Team (2016ZT06N331, 2017ZT07Z066), Guangdong Collaborative Innovation Center for Earthquake Prevention and Mitigation (2018B020207011), and Natural Science Foundation of Guangdong Province (2022A1515011105).

**Data Availability Statement:** Data sharing is not applicable to this article.

**Acknowledgments:** The raw tele-seismic waveforms in this study were provided by the Guangdong Seismic Network Center of the Guangdong Earthquake Agency. Some figures were plotted using Generic Mapping Tools (Wessel and Smith, 1995).

**Conflicts of Interest:** The authors declare no conflict of interest.

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
