# Peer review of "Shallow Crustal Structure of S-Wave Velocities in the Coastal Area of South China Constrained by Receiver Function Amplitudes"

_remotesensing, doi:10.3390/rs14122760_

Round 1

Reviewer 1 Report

Dear Authors

Your research is very important, useful, and complex to understand the behavior of certain natural phenomena in coastal sites. You start with mathematical analysis, then verify it, likewise with the rest.

These with my recommendations:

  • Improve the explanation of the used methodology, more detailed.
  • In Section: 2.4. Data Processing Procedures, you indicate that this process has 5 steps, but you only explain 4 of them.
  • Explain or describe the data structure of the stations. 
  • The conclusions are fine, reorganize them: after the specific conclusions, edit the final or general conclusion, to close the study objective. 

Author Response

Dear reviewer,

Thank you for reviewing our manuscript. We made great efforts to respond to all your comments during the revision and hope it will satisfy you. Details of revision please see the attachment.

Authors.

Reviewer 2 Report

The manuscript presents the shallow crustal Vs structure in the coastal
area of South China obtained by the inversion of the amplitude of receiver function of a selection of teleseismic events. They applied a recent and interesting development of receiver function methodology. I think that the manuscript should be improved before it can be accepted.

My main suggestions are:

  • improve the method description (relation between Qian method and Wang method, how do you convert alpha in depth? how do you convert optimized Vs to Vs of some specific depth?);
    re-organize some part of the text (description of geological characteristics of the area should be in Introduction, not in Data and Methods section; interpretations should be in Discussion section not in Results section);
  • revised the figures that show the results in order to be more clear.

Specific comments:

  • page 1, line 17:: “CASC” not yet defined
  • page 2, lines 54-55: references 18 and 19 used wave amplitude, not arrival time
  • page 2, lines 58-59: differences between Qian method and method in references 18 and 19
  • page 2, lines 72-81: I think that this part should be in “Introduction”, not in”Data and Methods”
  • page 2, lines 72-82: which kind of seismic stations?
  • page 2, lines 87-91: I think that this part should be in “Introduction”, not in”Data and Methods”
  • page 3, lines 102-103: “As the seismic wave arrives at the seismic station”: which kind of seismic wave?
  • page 3, lines 114: “dieletric”?
  • page 4, lines 140-142: equation 9 didn’t show “free surface displacement” as expected from the statement
  • page 4, lines 142: I don’t understand you write the equation in this form: the denominator should be square of eta.
  • page 4: lines 146: why “low frequency band”?
  • page 4, lines 149-150: “ray parameter (P)” -> “ray parameter (p)”?
  • page 5, lines 192-194: it is better if you explain more the results of Wang et al. (2018): the paper is in chinese and therefore not completely accessible to international readers.
  • page 5, lines 195-198: why did you not discuss your tests in terms of cut-off frequency that has a clear physical meaning?
  • page 6, lines 202: I am a little confused by your definition of mean square error.
  • page 6, figure 3a-3d: all your models show increase of Vs with depth, what happens with a low velocity channel?
  • page 6, figure 3e: how do you compute the depth of figure 3? Maybe the relation is explained in some referenced papers but it is a critical point and should be clarified. If I'm not wrong, the optimum value of Vs is the mean velocity up to some depth: how do you obtain the Vs at some specific depth?
  • page 7, line 223: “deburring” -> deblurring?
  • page 7, lines 225-229: did you use all the reported alpha values for each station and signal? It is not clear from your text.
  • page 7, line 270: “figure 2”-> figure4?
  • page 7, figure 4 : which alpha is used?
  • page 8, lines 263-268: I think that this part should be in Discussion
  • page 8, line 266: “dynamic model”?
  • page 8, figure 6: from this figure (and the others with the same color scale) I can distinguish only between values greater than or smaller than 3 km/s and I can’t find the difference between areas reported in Results section.
  • page 8, figure 10: why “. The two colored bars in (a) are inverted, i.e., the higher the color contrast is, the closer the result.”? In this way is harder to understand the figure
  • page 10, lines 314-315: “Geological studies” -> missing references.
  • page 14, line 441: “Data Availability Statement: Not applicable.” I think that this statement can be used only for theoretical studies, when no data was used. It is not your case.

Author Response

Dear reviewer, 

Thank you for reviewing our manuscript. We made great efforts to respond to all your comments during the revision. Details of revision please see the attachment.

authors.

Reviewer 3 Report

The article addresses an important and very interesting topic of the shallow crustal structure of S-wave velocities in the coastal area of South China constrained by receiver function amplitudes, which is appreciated. The study includes the numerical and theoretical research. In this paper the research is about establish a model for the evolution of the structural environment. The Reviewer has some concerns regarding the introduction, description of numerical models, description of results, conclusions and references. Generally, in this paper the English language should be improved by Native Speaker, because some sentence should be more clear. In opinion of Reviewer this paper should be subjected to major revision.

Other comments:

  1. Please explain more clear what differences are between your research and previous research cited in the text (more detailed)? In addition, please explain what is novelty of your research? Generally, this introduction is too short. In my opinion in this part of the text the Authors should be shown the impact of the seismic wave (earthquake) on civil engineering. In addition, please add the most popular methods use in civil engineering. These aspects are very important and in opinion of Reviewer should be improved. Below you can find a few papers about seismic response of civil engineering structures:
  • https://doi.org/10.1016/j.tust.2020.103808
  • https://doi.org/10.1016/j.soildyn.2021.107005
  • https://doi.org/10.1016/j.soildyn.2020.106484
  1. Detailed description of numerical model (use analysis, which program was used, boundary condition etc.) should be added.
  2. Page 3 (line 92 – 99) this is the description of Figure 1? Is too long. Please improve it.
  3. Please add the description of the lines on the Figure 3. Please improve it.
  4. Please improve the conclusions, because in current version are poor. What is the general conclusion from this research? What is the next step of your research? Please explain the tendency from your research.
  5. The references are unsatisfactory. Please add some papers of similar research, where the authors will be come from other country as China. Please improve it.

And the end I hope that my comments will be helpful for the authors.

Author Response

Dear reviewer,

Thank you for reviewing our manuscript. We made great efforts to respond to all your comments during the revision.

Details of revision please see the attachment.

Authors.

Round 2

Reviewer 1 Report

Dear

It is much better. 

Author Response

Thank you again for reviewing our manuscript. 

Reviewer 2 Report

The authors improved their manuscript and I think that there are only some minor problems. 

Specific comments:

page 2, lines 59-60: in your reply to my comments, you wrote “The author of this paper, Qian Y. and Shen X. Z.(2018), firstly proposed this idea of data processing (Qian et al., 2018), and Wang X. (Wang et al., 2019) later developed its application range and forward and inverse process.”, why do you not include this information in the manuscript?

page 7, lines 258-259: I agree with your description of the gaussian filter, but the point that I raised in the first revision is not about the choice of the filter itself, but about the use of “alpha” for the discussion of the tests. You can express the same gaussian filter in terms of cut-off frequency and the frequency is a physical quantities that has a clear impact on seismic wave. 

page 7, lines 264, equation 11: I don’t understand why in the formula there is a sum between the inverse of N and the other term.

There are some repetition in the pdf of the manuscript (e.g.: in abstract, lines 17-18 “the coastal area of South China (CASC)the coastal area of South China (CASC)the 17 CASC”): please check them.

Author Response

Dear reviewer,

Thank you for reviewing our manuscript again. Your comments on the details are very useful for us to revise the manuscript. We made great efforts to respond to all your comments during the revision. Details of revision please see the attachment.

Zhang Xin.

Reviewer 3 Report

Thank you for your improving.

Author Response

(The authors gave the same response as above.)
